# Impacts of Digital Information Management Systems on Green Transformation of Manufacturing Enterprises

**DOI:** 10.3390/ijerph20031840

**Published:** 2023-01-19

**Authors:** Zeyan Miao, Guohao Zhao

**Affiliations:** School of Business Administration, Shanxi University of Finance and Economics, Taiyuan 030006, China

**Keywords:** digital information management system, manufacturing enterprises, green transformation of enterprises, PSM-DID

## Abstract

Under the background of increasingly prominent environmental problems, the establishment and application of digital information management systems established by the digital economy have brought new opportunities and challenges to the green transformation of manufacturing enterprises. Considering the micro level of Chinese manufacturing enterprises, does the adoption of information management systems really promote the improvement of the green transformation level of manufacturing enterprises? This paper takes the adoption of digital information management systems by China’s enterprises as a “quasi natural” experiment and uses the Difference-in-Difference and Propensity Score Matching model (PSM-DID) to explore its impact on the green transformation of manufacturing enterprises and its mechanism. The study found that the adoption of digital information management systems by enterprises significantly improved the green transformation level of manufacturing enterprises, especially the symbolic green transformation level, and had no significant positive effect on the substantive green transformation level. The mechanism analysis shows that manufacturing enterprises can achieve green transformation by adopting information management systems to improve the digital level, strengthen green innovation ability, and increase the redundant resources of enterprises. The heterogeneity analysis based on the internal governance and external environment of enterprises shows that the adoption of digital information management systems by manufacturing enterprises has significantly improved the green transformation level of non-state-owned enterprises, enterprises with high corporate governance, non-heavily polluting enterprises, and enterprises in the eastern region. The research conclusion enriches the research related to digitalization and green transformation of enterprises and has important inspiration for Chinese manufacturing enterprises to use digitalization capabilities to seek green sustainable development under the wave of digital economy development.

## 1. Introduction

China’s “Fourteenth Five Year Plan” and the Outline of 2035 Long term Goals point out that we should actively promote the transformation and upgrading of traditional industries enabled by the digital economy to accelerate the low-carbon and green transformation of the development mode to promote the high-quality coordinated development of China’s economy, society, and ecological civilization. Furthermore, digital economy and green development should be adopted as the “two wings” to provide new momentum for high-quality development. As the material foundation and pillar industry of China’s national economy, manufacturing industry has made great contributions to GDP. According to the data of the *China National Bureau of Statistics* in 2021, the total number of manufacturing enterprises in China exceeds 3.5 million, accounting for more than 40% of the total number of enterprises in China. The added value of the manufacturing industry also increased from CNY 16.98 trillion in 2012 to CNY 31.4 trillion in 2021, accounting for 27.4% of the total national economy. The large base, rapid growth, and traditional extensive development mode place the manufacturing industry in the difficult situation of high energy consumption, high emissions, and high pollution that poses a great long-term threat to the ecological environment, while also rapidly improving the social economy and creating huge wealth [1]. Digitalization is an important means to promote the traditional manufacturing industry and realize its green transformation. The *SMARTer 2030* report released by *GESI* shows that digital technology can reduce global emissions by 20% in the next decade by enabling intelligent manufacturing, smart energy, and other industries. As an important implementation subject and micro foundation of green production, manufacturing enterprises urgently need to optimize production processes, upgrade the industrial structure, and optimize environmental governance through digital transformation. Digitalization has become the key link and core meaning of green, sustainable, and high-quality development of manufacturing enterprises.

At this stage, Chinese manufacturing enterprises have a poor level of initiative in green transformation, and the driving force of transformation mainly depends on external pressure to promote enterprises to achieve green transformation [2,3]. The survival of enterprises depends on their institutional environment, and the government, as the main determinant of the institutional environment, has a great influence on the green responsibility of enterprises. The implementation of the government’s macro-level digital policy at the enterprise micro level is characterized by the adoption of digital information management systems such as Enterprise Resource Planning (ERP), Manufacturing Execution System (MES), and Process Control System (PCS). The existing research believes that enterprises using digital information management systems can enhance the sustainable competitiveness of enterprises and achieve high-quality development of enterprises [4]. Specifically, it can not only reorganize business processes within the enterprise, provide support for management’s business decisions, reduce product costs and energy consumption, shorten product design and experiment cycles, improve customer satisfaction, and improve business efficiency, but also strengthen the enterprise’s ability to respond quickly to market changes externally [5]. At present, the development of China’s manufacturing industry is constrained by energy and resources, and the task of green development is arduous. Digitalization is an important means to transform the traditional manufacturing industry to achieve green transformation. [6]. On the other hand, there are still few theoretical research results on digital policy at present. A few of the existing policy studies mainly discuss emerging industrial policies at the macro level, summarize the experience of policy incentives in the development of intelligent manufacturing in other countries, and conduct qualitative research on the implementation effect of China’s existing policies at the macro and meso levels. The implementation of digitalization policy in manufacturing enterprises is embodied in the adoption of enterprise resource planning (ERP), manufacturing execution system (MES), process control system (PCS), and other information management systems. However, the existing literature does not cover the environmental effects and mechanisms that result from the adoption of information systems by enterprises.

Does the adoption of a digital information management system really promote the improvement of the green transformation level of manufacturing enterprises? If there is a driving effect, does it promote substantive green transformation or symbolic green transformation? What is the process mechanism to promote the green transformation level of enterprises? The existing research has not given a clear explanation of the above problems. In view of this, this paper focuses on the practical mapping of digital policies at the micro level and uses the Difference-in-Difference and Propensity Score Matching model (PSM-DID) to evaluate the effect of enterprises’ adoption of digital information management systems from the perspective of green transformation of manufacturing enterprises. Then, this paper further analyzes the internal mechanism of enterprises’ adoption of digital information management systems on green transformation of manufacturing enterprises, and establishes whether different internal governance and external environments of enterprises have a heterogeneous effect on this impact. Based on the perspective of implementing digital policies to micro enterprises, this paper pioneered the adoption of information management systems by manufacturing enterprises as a quasi natural experiment, and used the Difference-in-Difference and Propensity Score Matching model (PSM-DID) to explore the impact of digital policies on enterprise green transformation, enterprise substantive green transformation, and symbolic green transformation. This paper revealed the mechanism of digital policies on enterprise green transformation and enriched the relevant literature to test the effectiveness of digital policies from a micro perspective. The conclusion not only enriches the existing research on digitalization and green transformation of enterprises, but also has important enlightenment significance for Chinese manufacturing enterprises to use digitalization capabilities to seek green sustainable development under the wave of digital economy development.

## 2. Theoretical Analysis and Research Hypothesis

At the enterprise level, digitalization not only affects the supply chain, R&D, production, sales, and other production and operation processes and enterprise performance levels, but also affects corporate governance [7]. Under the condition of digitalization, the adoption of digital information management systems brings new management modes to manufacturing enterprises [8]. The digital information management system model of manufacturing enterprises is shown in Figure 1. Enterprise Resource Planning (ERP) advocates for the integration of internal and external resources of the enterprise. With financial management, production plan management, project management, and supply chain management as the core, ERP closely combines human, financial, material, production, supply, marketing, material flow, information flow, and management flow, and emphasizes the optimization and sharing of internal overall planning and overall resources of the enterprise. As the core of a digital factory, the Manufacturing Execution System (MES) plays an important role in manufacturing enterprises. It focuses on basic factory information, and lean production management. The Laboratory Information Management System (LIMS) emphasizes the optimal control of the production process, shares ERP information with the enterprise, and achieves a high degree of integration of enterprise production data and information applications. In this way, it can effectively promote enterprises to carry out fine production, enhancing the competitiveness of enterprises. The Process Control System (PCS) focuses on real-time optimal control, basic control, and advanced control, and emphasizes process optimization. The layers in the three-layer digital information management system of manufacturing enterprises complement each other. Through the adoption of the digital information management system, manufacturing enterprises can optimize the allocation of internal and external resources, promote enterprises to carry out refined production, optimize production process control, improve supply chain and customer relationship management, and achieve integrated management and control of internal business and information integration and sharing in order to meet the information needs of manufacturing enterprises required to maintain competitive advantage and sustainable development.

The core of the digital information management system adopted by manufacturing enterprises is the enterprise’s digital strategic change. Enterprise digitalization is the integration of digital technology and enterprise business processes [9] that improve enterprise value, competitiveness, and influence through the use of digital technology [10]. Modern manufacturing enterprises are driven by the adoption of digital information management systems and innovation management, and the two are integrated to form a “dynamic core capability of digital information management”. According to the theory of dynamic capabilities, dynamic capabilities promote enterprises to maintain and update existing resources through the ability to perceive and control opportunities and restructure strategies so as to quickly respond to and meet the requirements of green and low-carbon development in the outside world and achieve green transformation [11]. This paper suggests that the impact of digital information management systems on the green transformation of enterprises by improving the degree of digitalization of manufacturing enterprises is mainly based on the following three aspects. First of all, based on the ability of opportunity perception, manufacturing enterprises can timely scan the external environment of the enterprise to identify possible opportunities and threats, integrate and integrate data information, form digital thinking, and accurately predict and quickly respond to external green and low-carbon development needs by using digital information management systems. Secondly, based on the ability to control opportunities, manufacturing enterprises firmly seize the opportunities of the “green low-carbon” and “digital” era by adopting digital information management systems that provide a forward-looking advantage, effectively conduct dynamic monitoring on internal research and development, production, sales, etc., and accelerate the integration of internal and external knowledge of enterprises, especially in the fields of energy conservation, emission reduction, and environmental protection, so as to promote green transformation. Finally, based on the ability of strategic reconstruction, in the context of achieving China’s “Strive to reach the peak of carbon dioxide emissions by 2030, and strive to achieve carbon neutrality by 2060” goal, manufacturing enterprises regard green transformation as an important strategic measure for sustainable development. On the one hand, manufacturing enterprises use digital information management systems to analyze, utilize, and process the obtained digital information in order to embed data information into enterprise economic activities, business activities, and management activities; monitor the resource and energy situation from the whole link and process of the enterprise; reduce enterprise carbon emissions on the basis of improving resource and energy utilization efficiency; and achieve green transformation of the enterprise. On the other hand, manufacturing enterprises can also strengthen the connection between enterprises in the supply chain through the use of digital information management systems, which is conducive to the formation of a new pattern of openness, collaboration, and sharing in the manufacturing industry, promoting resource sharing in all sectors of the industry and the supply chain, reducing repeated construction and investment waste, creating a green supply chain, realizing green transformation of enterprises in the chain, and making green transformation of manufacturing enterprises easier to achieve. Therefore, the following assumptions are made.

**Hypothesis** **1.**
*Digital information management systems can improve the degree of digitalization of manufacturing enterprises, thereby improving the level of green transformation of enterprises.*


Green innovation refers to new products, services, processes, or management systems used by enterprises to deal with environmental problems [12]. At present, academia divides green innovation into green technology innovation and green management innovation [13]. Green technology innovation aims to integrate environmental knowledge and technology. Manufacturing enterprises introduce new products and technologies or improve existing products or processes through green technology innovation to save energy and resources, promote harmonious development between environment, economy and production, and realize green transformation of enterprises [14]. Green management innovation refers to the integration of the “green” concept into the entire production and operation process of the enterprise, which not only includes the greening of corporate culture and behavior, but also includes the innovation of green cost management, green marketing, green supply chain management, etc., in the business process [15]. This paper suggests that the impact of digital information management systems on the green transformation of enterprises by promoting the green innovation capability of manufacturing enterprises is mainly based on the following two aspects. On the one hand, from the perspective of green technology innovation, on the other hand, the resource-based theory believes that the technological capability of an enterprise is the basis for its green technology innovation, so as to build a sustainable competitive advantage [16]. The adoption of digital information management systems by a manufacturing enterprise is an important embodiment of an enterprise’s technological capabilities. Second, the adoption of digital information management systems by manufacturing enterprises can stimulate enterprises to carry out joint green innovation with other enterprises, promoting knowledge sharing [17] and improving their green independent technological innovation capability based on existing technologies. The development, operation, and innovation of green technology can effectively reduce the discharge of wastewater, waste gas, and solid waste of enterprises, and the emergence of green technology innovation experiences a curve effect, which makes the optimization of resource allocation, improvement of resource and energy utilization efficiency, and pollution control and emission effects brought by green technology capabilities gradually appear, thus contributing to the realization of the green transformation of manufacturing enterprises [18]. On the other hand, from the perspective of green management innovation, manufacturing enterprises can not only achieve paperless office work within the enterprise, reduce resource consumption, and save enterprise costs, but also achieve an effective link from raw material procurement to after-sales service in the whole manufacturing process by adopting digital information management systems. This greatly improves the timeliness and efficiency of each functional link of the production enterprise, thus improving the enterprise’s green management innovation capability. Therefore, the following assumptions are made.

**Hypothesis** **2.**
*Digital information management systems can strengthen the green innovation ability of manufacturing enterprises and then improve the level of green transformation.*


Due to the limited rationality of enterprise strategic decision making, redundant resources are very common in the operation of enterprises, which has attracted the attention of domestic and foreign strategic management scholars. Redundant resources are defined as the minimum resources owned by an enterprise that exceed its own survival requirements [19]. Within an enterprise, redundant resources can serve as a buffer to enhance the flexibility of the enterprise. According to the resource allocation theory, the key for an enterprise to obtain sustainable competitive advantage is how to effectively allocate and sustainably utilize its redundant resources or core resources. This paper believes that the impact of digital information management systems on the green transformation of enterprises by increasing redundant resources of manufacturing enterprises is mainly based on the following three aspects. First of all, from the perspective of production management, manufacturing enterprises can effectively strengthen the dynamic control and management of the production line by adopting ERP, MES, PCS, and other digital information management systems, using the technology of the Internet of things and equipment monitoring technology, making use of intelligent analysis systems to develop green production processes, and reasonably arranging production plans and scheduling production progress, so as to fully reduce the energy consumption and resource waste required in the production process. At this time, the redundant resources of enterprises increase and the flexibility of enterprises is enhanced. Second, from the perspective of warehousing management, manufacturing enterprises, through the use of digital information management systems, reasonably set up warehousing networks and facilities to store materials according to the characteristics and requirements of goods, and to carry out storage, loading and unloading, and handling. At the same time, materials are coded to achieve batch and shelf life business management of materials, reduce damage to goods caused by inventory errors, and avoid resource waste. At this time, redundant resources of enterprises increase. Third, from the perspective of logistics management, manufacturing enterprises can not only reasonably plan and implement logistics activities such as storage, packaging, transportation, loading and unloading, handling, distribution, and information processing, but also use the wastes generated in production in order to achieve remanufacturing, recycling, and greening through the use of digital information management systems. At this time, enterprises have more redundant resources. Redundant resources can ease the cost pressure caused by the increase in environmental protection investment, and help enterprises increase resource investment in green technology innovation [20]. This effectively buffers the additional costs caused by green technology innovation and the pressure of resource shortage in the process of green transformation of manufacturing enterprises. Furthermore, the existence of redundant resources will have a positive impact on the development of green product innovation, making the green transformation of manufacturing enterprises easier to achieve [21]. Therefore, the following assumptions are made.

**Hypothesis** **3.**
*Digital information management systems can increase the redundant resources of manufacturing enterprises, thereby improving the level of green transformation of enterprises.*


Based on the above analysis, the basic inference of this paper is that manufacturing enterprises can improve their green transformation level by adopting digital information management systems.

## 3. Research Design

### 3.1. Data Sources

With the popularization of digital information technology in 2011, ERP and other digital management information systems were widely used in manufacturing enterprises, laying a technical foundation for the realization of digital transformation of enterprises. Due to the impact of COVID-19 in 2020, the financial data of enterprises have undergone great changes. Therefore, this paper selects the data of listed A-share manufacturing companies from 2011 to 2019 as the initial sample. The data of the digital information management system used by the enterprise in this paper are collected manually, and other relevant data are from the CSMAR and Wind databases. In addition, this paper excludes companies with incomplete annual report data, ST and * ST enterprise samples, enterprises that have adopted digital information management systems, and enterprises with missing data in 2011. The observed values were winsorized at 1% and 99% to eliminate the impact of outliers on the research conclusions.

### 3.2. Model and Variables

#### 3.2.1. Model Specification

The Difference-in-Difference model (DID) is widely used in the field of policy effects. The usual practice is to set up an experimental group and a control group, and then use the control group that is not affected by the policy to investigate the “counterfactual” situation of the experimental group if it is not affected by the policy so as to measure the implementation effect of the policy [22]. This paper focuses on the impact of digital policies on the green transformation of manufacturing enterprises, and the implementation of digital policies at the micro level of enterprises is characterized by the adoption of information management systems such as ERP, MES, and PCS. Therefore, using the ideas of experimental economics for reference, we can regard the adoption of information management systems by enterprises as a “quasi natural” experiment, and estimate the green spillover effect of the adoption of digital information management systems by examining the difference in the level of green transformation between manufacturing enterprises (experimental groups) that adopt digital information management systems and enterprises (control groups) that do not. However, if the comparison between the experimental group and the control group is conducted directly, it may not only lead to the problem of sample selection error, i.e., endogeneity, but may also be interfered with by the unobservable factors and the inherent differences between individuals that do not change over time, which may lead to the deviation of the main effect estimation results. The tendency score matching method (PSM) proposed by Rosenbaum and Rubbin (1983) can solve this problem well [23]. PSM is a kind of statistical method that uses non-experimental data or observational data to analyze intervention effects. The theoretical framework of propensity score matching is “counterfactual inference model”. The “counterfactual inference model” assumes that any research object of causal analysis has results under two conditions: observed and unobserved. The basic idea is to take manufacturing enterprises adopting digital information management systems as the experimental group, calculate the probability value of enterprises adopting digital information management systems according to a set of matching variables, and then determine the same or similar samples from the sample group of manufacturing enterprises not adopting digital information management systems as the control group. In this way, the biggest difference between the matched experimental group and the control group is whether the digital information management system is used to eliminate the selective deviation of samples and solve the “counterfactual” problem.

This paper adopts the Difference-in-Difference (DID) method to eliminate the unobservable factors that change with time and the inherent differences that do not change with time among individual enterprises through twice difference, and estimates the green spillover effect of manufacturing enterprises using digital information management systems. Therefore, this paper builds the following model based on the practices of Beck et al. (2010) [24]:(1)$$GT_{i,t}=\alpha_{0}+\alpha_{1}Treat_{i}\times Post_{it}+\beta controls_{it}+\mu_{i}+\theta_{t}+\varepsilon_{i,t}$$
where i is the enterprise, t is the time, and $GT_{i,t}$ is the explanatory variable, which indicates the green transformation level of manufacturing enterprises i in period t. $Treat_{i}$ is the grouping variables of digital information management systems for manufacturing enterprises. $Post_{it}$ is the period variables for enterprises i to adopt digital information management systems in the t period. $X_{i}$ is the control variable, $\mu_{i}$ is the individual fixed effect of enterprises; $\theta_{t}$ is the year fixed effect; and $\varepsilon_{i,t}$ is a random disturbance term.

According to the above model, the green transformation levels of manufacturing enterprises ($Treat_{i}$ = 1) adopting digital information management systems before and after the adoption are $\alpha_{0}$ +$\;\alpha_{2}$ and $\alpha_{0}$ +$\;\alpha_{1}$ +$\;\alpha_{2}$ +$\;\alpha_{3}$. The difference is $\alpha_{1}$ +$\;\alpha_{3}$. Manufacturing enterprises ($Treat_{i}$ = 0) that do not use digital information management systems and the enterprises that do use digital information management systems with a difference of $\alpha_{1}$ are equal to the coefficient term of *Post* × *Treat*. This is the net green spillover effect of adopting digital information management systems for enterprises.

#### 3.2.2. Variable Introduction

Green transformation of enterprises (*GT*). The data of enterprises’ green transformation were based on the research of Li Weian et al. (2019) [25] and is mapped to the micro enterprise level according to the core requirements of green development “source reduction, process control, and end governance”, to measure the enterprises’ green low-carbon transformation. Specifically, the value is assigned according to the green low-carbon transformation performance of enterprises in these three dimensions. For the non-quantifiable indicator, if there is such advantage, it is assigned a value of 1; otherwise, it is 0. For quantifiable indicators, it is 1 if there is a written description, 2 if there is a numerical description, and 0 if there is no description. Finally, add all scores to obtain the total score of the green low-carbon transformation of enterprises. In addition, according to the traditional social responsibility theory, there are mainly two types of motivations for enterprises to undertake social responsibility: altruistic motivation and instrumental motivation [26,27]. Two different opportunities lead to different results. Accordingly, this paper divides the green transformation of manufacturing enterprises into substantive green transformation and symbolic green transformation. The former pursues the substantive significance of green transformation, that is, by taking multiple substantive ways to obtain source control, process emission reduction, and end-of-pipe governance to achieve the green transformation of enterprises, while the latter is a behavioral choice driven by symbolic significance. The specific measurement formula is as follows:Enterprise green transformation = Source reduction + Process control + End treatment = (Wastewater discharge + COD discharge + $SO_{2}$ Discharge + $CO_{2}$ discharge + Smoke and dust discharge + Industrial solid waste generation) + (Environmental information disclosure + Social responsibility report + Environmental report + Whether to pass ISO14001 certification + Environmental protection investment + Environmental protection concept + Environmental protection objectives + Environmental petition cases + Environmental management system + Environmental violations + “Three simultaneities” System + Environmental education and training + Special action for environmental protection + Key pollution monitoring units + Emergency response mechanism for environmental events + Pollutant discharge up to standard + Environmental honor or reward + Sudden environmental accident + Whether ISO9001 certification has been passed) + (Waste gas emission reduction + Waste water emission reduction + Dust, smoke and dust control + Solid waste utilization and disposal + noise, light pollution, radiation and other control + Cleaner production implementation)Substantial green transformation = Wastewater discharge + COD discharge + $SO_{2}$ discharge + $CO_{2}$ discharge + Smoke and dust discharge + Industrial solid waste generation + Environmental protection investment + Environmental petition cases + Environmental violations + Special environmental protection actions + Key pollution monitoring units + Sudden environmental accidents + Cleaner production implementationSymbolic green transformation = Environmental information disclosure + Social responsibility report + Environmental report + Whether to pass ISO14001 certification + Environmental protection concept + Environmental protection goal + Environmental protection management system + “Three simultaneities” system + Environmental protection education and training + Environmental event emergency mechanism + Pollutant discharge standard + Environmental protection honor or reward + Whether to pass ISO9001 certification + Waste gas emission reduction treatment + Waste water emission reduction treatment + Dust, smoke and dust treatment + Solid waste utilization and disposal + Noise, light pollution Radiation treatment.

Digital information management system (*Post* × *Treat*). Because the research adopts the Difference-in-Difference model (DID), the grouping variable *Treat* is set. The group of digital information management system adopted by the enterprise is 1, and the group not adopted is 0. Set the period variable *Post*. The period before the enterprise adopts the digital information management system is 0, and the period after the adoption is 1. The net green spillover effect of enterprises adopting digital information management systems is the interaction item between *Post* and *Treat* (*Post* × *Treat*). The data of the digital information management system adopted by the enterprise were collected manually as referred to by Dorantes et al. (2013) [28]. The specific steps are as follows: (1) Download the annual reports of all Chinese A-share-listed manufacturing companies from 2011 to 2019 through http://www.cninfo.com.cn/new/index (accessed on 1 November 2022), and use Python to create “ERP”, “MES”, “PCS”, “Enterprise Resource Planning”, “Manufacturing Execution System”, “Process control system” and the names of digital information management system suppliers. The manufacturing companies adopting digital information management systems are preliminarily selected. (2) Read the selected annual reports one by one, and determine the time when the implementation of the enterprise digital information management system is completed or officially used. Relevant keywords include “online”, “startup”, “completion”, “implementation”, “reference”, “put into operation”, etc. (3) For the enterprise that has not identified the implementation of a digital information management system in the annual report of the selected year, expand the search scope to all announcements of the enterprise since its listing, and repeat step (2). (4) Conduct a Baidu search for enterprises that have not identified the use of digital information management systems in the annual report of years outside the selected range. (5) Verify the filtered information with the information on the official website of the enterprise or the searchable Baidu search results to ensure the accuracy of the data used by the enterprise in the digital information management system.

Control variable. Referring to the existing research, this paper controls the relevant variables at the enterprise level, the region where the enterprise is located, and the industry level. The Size of the enterprise (*Size*) is expressed by the natural logarithm of the total assets of the enterprise. Age of the enterprise (*Age*) is measured by the natural logarithm of the number of years from the date of establishment to the observation period. RD investment (*Rdv*) takes the natural logarithm of the RD investment amount. Asset liability ratio (*Lev*) is expressed by the ratio of total liabilities at the end of the year to total assets at the end of the year. Return on total assets (*ROA*) is expressed as the ratio of the enterprise’s net profit to the average balance of total assets. Enterprise growth (*Growth*) is the operating profit growth rate. It is expressed by the amount of operating profit in the current period of this year—the amount of operating profit in the same period of last year/the amount of operating profit in the same period of last year. The board size (*Board*) is expressed as the natural logarithm of the total number of directors. In addition, this paper also controls the marketization level (*Market*), industry level industry concentration (*HHI*), individual fixed effect of enterprises, and year fixed effect of the region where the enterprise is located.

Table 1 reports the descriptive statistical results of all variables in the full sample. The results show that the average green transformation level of Chinese manufacturing enterprises is 6.68, the maximum value is 28, the minimum value is 1, and the standard deviation is 6.53, indicating that the overall green transformation level of Chinese manufacturing enterprises is still low, and there are still large differences among individual enterprises. The average level of symbolic green transformation is 4.76, and the average level of substantive green transformation is 1.91, indicating that most Chinese manufacturing enterprises are still at the level of symbolic green transformation, and there is still much room for improvement in terms of substantive green transformation.

## 4. Empirical Analysis

### 4.1. Propensity Score Matching (PSM)

The premise of using the Difference-in-Difference model (DID) for policy evaluation is to meet the parallel trend assumption, while the Propensity Score Matching (PSM) method can solve the problem of sample selection bias, thus meeting the requirements of the DID model for common trends. First of all, on the basis of existing research, this paper selects enterprise size, enterprise age, RD investment, asset-liability ratio, return on total assets, enterprise growth, board size, marketization level, and industry concentration as matching estimators to calculate the propensity score. On the basis of the propensity score, we used StataMP17 software to perform 1:3 nearest neighbor matching. The balance test results are shown in Table 2. After the propensity score was matched, the difference between the experimental group and the control group on each matching variable was significantly reduced, the absolute value of the standardized deviation (% bias) of all variables was less than 10%, and the *t*-test results showed that there was no significant statistical difference between the experimental group and the control group on the observable variables. Therefore, the matching effect is good.

In addition, the comparison of the standard deviation before and after the tendency score matching is shown in Figure 2. The standard deviation after matching is distributed around 0, which is significantly smaller. This shows that the deviation between the variables after matching becomes smaller, and the experimental group and the control group are more similar after the sample is matched for the propensity score. Furthermore, investigation of whether there are differences between the two groups’ propensity scores before and after matching was conducted. According to Figure 3, the two nuclear density curves of the experimental group and the control group before and after matching have large deviations. After matching, the mean line of the sample propensity scores of the experimental group is closer to the mean line of the control group. Therefore, the matching effect is good to some extent.

### 4.2. Analysis of Difference-in-Difference Model Results

Based on the PSM method, the scores of the experimental group and the control group were matched, and a new experimental group and the control group were further obtained. The Difference-in-Difference model was used to estimate the new samples. Considering that the matched control group sample may be the matching object of multiple processing group samples, the importance of control group samples with different weights in the overall control group samples is different. The larger the weight is, the more times matched, and the more attention should be paid to participating in regression. Therefore, a feasible method is to copy the matched samples in the control group according to the weight, which is called frequency-weighted regression [29]. The results are shown in Table 3. Model (1) is regression without control variables, and models (2), (3), and (4) are regressed by adding control variables. In models (1) and (2), the coefficients of *Post* × *Treat* are positive and significant, indicating that manufacturing enterprises have significantly improved the level of green transformation by adopting digital information management systems. Consistent with the basic inference of this paper, model (2) has not changed because of the addition of control variables, so it can be preliminarily considered that the results of DID are robust. Furthermore, the green transformation level is divided into the symbolic green transformation level and the substantive green transformation level; the coefficient of *Post* × *Treat* is positive and significant in model (3), but not significant in model (4). This indicates that the adoption of digital information management systems by Chinese manufacturing enterprises has significantly improved the level of symbolic green transformation, and the promotion effect on the level of substantive green transformation is not significant. This is consistent with the fact that Chinese manufacturing enterprises have less initiative to carry out green transformation at this stage, and the driving force of transformation mainly depends on external pressure to promote enterprises to achieve green transformation [2,3]. China’s manufacturing enterprises still have much room for improvement in terms of substantial green transformation.

### 4.3. Robustness Check

First, in order to reduce the estimation error caused by the selection of nearest neighbor matching proportion, this paper further adopts 1:2 nearest neighbor matching. Table 4 reports the matching balance test results of the 1:2 nearest neighbor matching propensity score. It can be seen that the matching result in Table 4 is similar to the previous trend score, and the matching effect is good. Then, we repeated the regression in Table 3 and compared it with the DID regression results mentioned above. The stability test results of DID regression in Table 5 show the regression results of model (1) without control variables and model (2) with control variables. The coefficients of *Post* × *Treat* are positive and significant. The coefficient of *Post* × *Treat* is positive and significant in model (3). The coefficient of *Post* × *Treat* is positive but not significant in model (4), which is consistent with the aforementioned double difference regression results, and the regression results are robust.

Second, change the research method. In the research on the green spillover effect of enterprises using digital information management systems, mixed OLS regression is used. The regression results are shown in Table 5, model (5). It shows the coefficient of *Post* × *Treat* is positive and significant, indicating that manufacturing enterprises have significantly improved the level of green transformation by adopting digital information management systems. The regression results are robust.

Finally, compared with the IT manufacturing industry, which is born with a certain digital gene, the traditional manufacturing industry needs to activate its potential through digitalization to achieve high-quality development [30]. Therefore, the regression is conducted again after excluding computer, communication, and other electronic equipment manufacturing industries. It shows the regression results are shown in Table 6, model (6). The coefficient of *Post* × *Treat* is still positive and significant, indicating that manufacturing enterprises have significantly improved the level of green transformation by adopting digital information management systems. The regression results are robust.

## 5. Further Analysis

### 5.1. The Internal Mechanism

From the above empirical analysis, it can be seen that manufacturing enterprises have significantly improved the level of green transformation by adopting digital information management systems. In order to further clarify the internal mechanism of enterprises’ adoption of digital information management systems affecting enterprises’ green innovation, based on the above theoretical analysis and research assumptions, this paper uses the three-step test method used by Baron and Kenny (1986) [31] to test the enterprises’ digital level (*Dig*) to determine whether green innovation capability (*GI*) and redundant resources (*RR*) have an intermediary effect between manufacturing enterprises’ adoption of digital information management systems and improvement of green transformation level. The specific model is as follows:(2)$$Mediation_{i,t}=\alpha_{0}+\alpha_{2}Treat_{i}\times Post_{it}+\beta controls_{it}+\mu_{i}+\theta_{t}+\varepsilon_{i,t}$$
(3)$$GT_{i,t}=\alpha_{0}+\alpha_{3}Treat_{i}\times Post_{it}+\alpha_{4}Mediation_{i,t}+\beta controls_{it}+\mu_{i}+\theta_{t}+\varepsilon_{i,t}$$
where $Mediation_{i,t}$ is the intermediary variables. They are the enterprises’ digital level (*Dig*), green innovation capability (*GI*), and redundant resources (*RR*). Other variables remain unchanged. 

First, the intermediary effect of enterprises’ digitization level (*Dig*) is tested. This paper uses Qi Huaijin et al. (2020) [32] research to measure the digital level of enterprises (*Dig*) by using the proportion of the digital-economy-related part of the year-end intangible asset details disclosed in the notes of the financial reports of listed companies in the total intangible assets [31]. The aforementioned DID regression results show that manufacturing enterprises significantly improved the level of green transformation by adopting digital information management systems. The coefficient is positive and significant. The coefficient of *Post* × *Treat* is significantly positive and the coefficient of *Dig* is significantly negative in model (2), Table 6. The intermediary effect of enterprises’ digital level (*Dig*) is established. This means that digital information management systems can improve the degree of digitalization of manufacturing enterprises and thus enhance the level of green transformation of enterprises. Therefore, we assume that Hypothesis 1 is true.

Secondly, test the intermediary effect of green innovation capability (*GI*). Based on the fact that manufacturing enterprises have significantly improved the level of green transformation by adopting digital information management systems, the model (3) in Table 6 shows that the coefficient of *Post* × *Treat* is positive and significant. In model (4), the coefficients of *Post* × *Treat* and *GI* are significantly positive. The intermediary effect of green innovation capability (*GI*) is established. That is, digital information management systems can strengthen the green innovation ability of manufacturing enterprises, and then enhance the level of green transformation of enterprises. Therefore, we suppose Hypothesis 2 holds.

Finally, the mediating effect of redundant resources (*RR*) is tested. This paper measures redundant resources by using the mean value of three dimensions: non-precipitation redundant resources, precipitation redundant resources, and potential redundant resources. The results are shown in Table 6, models (5) and (6). Model (5) shows that the coefficient of *Post* × *Treat* is positive and significant. The coefficients of the coefficients of *Post* × *Treat* and *RR* are significantly positive. The intermediary effect of redundant resources (*RR*) is established. That is, digital information management systems can increase the redundant resources of manufacturing enterprises, thereby improving the level of green transformation of enterprises. Therefore, we assume Hypothesis 3 holds.

### 5.2. Heterogeneity Analysis

The green transformation of enterprises is a change in their strategic positioning to achieve the dynamic balance between the internal governance of enterprises and the external environment of enterprises [33]. The above empirical results show that the adoption of digital information management systems by manufacturing enterprises has significantly improved the level of green transformation, so whether different internal corporate governance and external environment will have a heterogeneous effect on this impact is a question that needs further study. This paper analyzes the heterogeneity by introducing the internal governance variables, such as the nature of enterprise property rights, the level of enterprise internal governance, and the external environment variables, such as the characteristics of polluting industries and regional geographic location, through grouping regression.

First, based on the nature of enterprise property rights, the sample enterprises are divided into state-owned enterprises and non-state-owned enterprises for grouping regression. The research results are shown in Table 7, models (1) and (2). The coefficient of *Post* × *Treat* in non-state-owned enterprises is significantly positive at the level of 5%, while the state-owned enterprises have no significant impact. Therefore, the adoption of digital information management systems by manufacturing enterprises has significantly improved the green transformation level of non-state-owned enterprises, and the green spillover effect on state-owned enterprises is not significant. The main reason is that compared with state-owned enterprises, non-state-owned enterprises are facing the pressure of government environmental supervision for their sustainable development. In order to obtain institutional legitimacy, enterprises have a stronger motivation to carry out green transformation, receive less attention from the public and media supervision, and are more flexible in green transformation. In this case, it is more conducive for the digital information management system to play a role in the green transformation of enterprises.

Secondly, based on the perspective of corporate governance level, this paper classifies enterprises with corporate governance levels greater than the median of the sample enterprises into enterprises with high corporate governance levels, or enterprises with low corporate governance levels, and then conducts group regression. The research results are shown in Table 7, models (3) and (4). The coefficient of *Post* × *Treat* in enterprises with high corporate governance levels is significantly positive at 1%, while enterprises with low corporate governance levels have no significant impact. Therefore, the adoption of digital information management systems by manufacturing enterprises has significantly improved the green transformation levels of enterprises with high corporate governance levels, and the green spillover effect on enterprises with low corporate governance levels is not significant. The main reason is that, compared with enterprises with low corporate governance levels, enterprises with high corporate governance levels rely less on enterprise digital information management systems. When the level of corporate governance is low, the dependence on the enterprise’s digital information management system is high. At this time, the enterprise will embed the digital information management system as the main task. The complexity of the enterprise’s green product development, green technology innovation, green management innovation, and the probability of failure of green technology innovation are relatively high. Manufacturing enterprises will inevitably face higher learning costs and trial and error costs. In this process, enterprises will also face the situation of complex and interlaced external knowledge and internal knowledge, which will lead to enterprises paying more “tuition fees”, and will also lead to more waste of resources and energy and environmental pollution, which is not conducive to the green transformation of enterprises. When the level of corporate governance is high, it is easier for enterprises to achieve the effect of “1 + 1 > 2” by adopting digital information management systems for green transformation.

Thirdly, based on the characteristics of the pollution industry, this paper uses Luo W et al.’s (2022) [34] research for reference to divide the sample enterprises into heavy pollution enterprises and non-heavy polluting enterprises for grouping regression [33]. The research results are shown in Table 7, models (5) and (6). The coefficient of *Post* × *Treat* in non-heavily polluting enterprises is significantly positive at the level of 5% and the coefficient of *Post* × *Treat* in heavily polluting enterprises is significantly negative at the 1% level, indicating that the adoption of digital information management systems by manufacturing enterprises has significantly improved the green transformation level of non-heavily polluting enterprises but significantly reduced the green transformation level of heavily polluting enterprises. The main reason is that compared with heavily polluting enterprises, non-heavily polluting enterprises are less subject to government environmental policy regulation, social legitimacy pressure, and ecological environment threats. In this context, manufacturing enterprises have a greater choice to promote green transformation through digital information management systems.

## 6. Conclusions

This paper focuses on the practice of digital policy at the macro level in manufacturing enterprises at the micro level, and takes the adoption of digital information management systems by enterprises as a “quasi natural” experiment. With A-share-listed manufacturing enterprises as the research sample, and the research interval from 2011 to 2019, the final research sample is determined through the PSM-DID model to study the impact of digital information management systems on the green transformation level of manufacturing enterprises. The research finds that: first, the digital information management system significantly improved the green transformation level of manufacturing enterprises, especially the symbolic green transformation level, but has not yet produced significant positive effects on the substantive green transformation level. Second, the analysis of the mechanism shows that manufacturing enterprises can improve the level of green transformation by adopting information management systems because it improves the enterprises’ digital level, promoting green innovation ability, and increasing redundant resources of enterprises. Thirdly, the heterogeneity analysis based on the internal governance and external environment of enterprises shows that the adoption of digital information management systems by manufacturing enterprises significantly improved the green transformation level of non-state-owned enterprises, enterprises with high corporate governance levels, non-heavily polluting enterprises, and enterprises in the eastern region.

The research conclusion enriches the research related to digitalization and green transformation of enterprises and has important inspiration for Chinese manufacturing enterprises to use digitalization capabilities to seek green sustainable development under the wave of digital economy development. Moreover, this study has some implications for implementing and improving the practice of digital policies at the micro level of manufacturing enterprises and promoting the green transformation and high-quality development of manufacturing enterprises.

(1)The information inheritance and interaction of the whole manufacturing enterprise is realized on the basis of effectively linking various digital information management systems within the manufacturing enterprise. The Fourth Plenary Session of the 19th Central Committee of the People’s Republic of China formally proposed to juxtapose data, land, labor, capital, technology, etc., as production factors. The Fourteenth Five Year Digital Economy Development Plan issued by the State Council also emphasized that data factors are the core engine for deepening the development of the digital economy. At this stage, although most manufacturing enterprises have been proficient in using digital information management systems, traditional manufacturing enterprises still have problems such as isolated information islands, lack of top-level design, and vertical use of multiple applications. All digital information management systems have not been effectively connected. Although a large amount of data have been accumulated, it is difficult for manufacturing enterprises to produce and manage. The problem of information interaction in operation and other links makes it difficult to enhance the core value of the enterprise. In the process of implementing the digital transformation of enterprises, it is necessary to effectively link various digital information management systems within the manufacturing enterprises to realize the interconnection of data. Through the development of the enterprise’s internal management system, it is necessary to achieve the interconnection of external product data, so as to realize the timely interconnection of the enterprise and the industrial supply chain data, which greatly improves the industrial distribution and operation efficiency, and significantly reduces the enterprise’s operating costs.(2)Build green digital information management systems for manufacturing enterprises. The construction of green digital information management systems by manufacturing enterprises refers to the accounting of environmental investment and expenses generated by various activities in the operation and production process of manufacturing enterprises through the use of the prediction, planning, accounting, control, analysis, and other functions of the digital information management system, so as to achieve the purpose of controlling environmental costs. The reason why enterprises can continue to grow is that they need to innovate and make breakthroughs in business, management, and capability. By using digital information management systems, manufacturing enterprises can fundamentally promote the improvement of their business ability, so as to achieve the optimal business level and optimal business model. If manufacturing enterprises want to truly realize green transformation, they must first build a green digital information management system for manufacturing enterprises. Secondly, they must organically combine the green digital management system with the existing management links of enterprises. Finally, they should improve and reconstruct the enterprise culture and values from the aspects of improving the information management ability, operation management ability, and business ability of enterprises.(3)Introduce and cultivate digital informatization talents. In enterprises, from the top management level to the employees, digital thinking needs to be cultivated. As the “leader” of the enterprise, the enterprise executives can effectively promote the digital transformation in the enterprise only if they all recognize and believe in the benefits of digital transformation. All company executives should play the role of “facilitator” and help the whole company form the idea of digital transformation through persistent communication and constant preaching. Employees are the “screws” of enterprise operation. Whether they are downstream employees who fully implement system instructions or upstream employees who optimize system decisions, they should aim at compound talents who understand both technology and management. Armed with the dual means of technology and management, employees can protect the digital transformation of enterprises.(4)Explore the high-quality development path of promoting green transformation of Chinese manufacturing enterprises with digital technology. Today, sustainable development has become an important goal and challenge of global development. More and more enterprises are actively seeking a greener, more efficient, and more innovative development mode, and use digital empowerment to achieve a win–win situation for economic development and environmental protection. The manufacturing industry is one of the major industries of carbon emission in China. Promoting green transformation and upgrading manufacturing enterprises is the core of achieving sustainable socioeconomic development. In the face of the macro background of green, low-carbon, and sustainable development, domestic manufacturing enterprises must further strengthen production process control, carry out green technology innovation, use digital technologies such as enterprise digital information management systems to achieve the purpose of improving resource and energy utilization and energy conservation and emission reduction, and boost the realization of China’s “dual carbon” goal.

## Figures and Tables

**Figure 1 ijerph-20-01840-f001:**
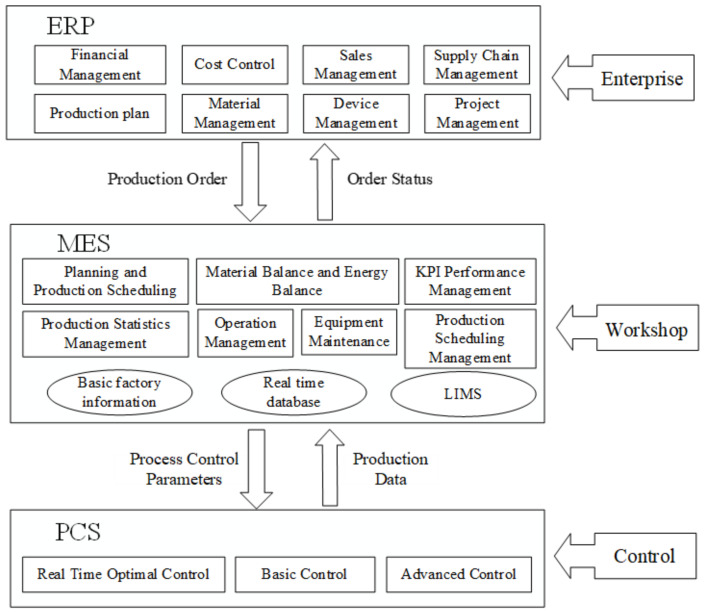
The Digital Information Management System Model of Manufacturing Enterprises.

**Figure 2 ijerph-20-01840-f002:**
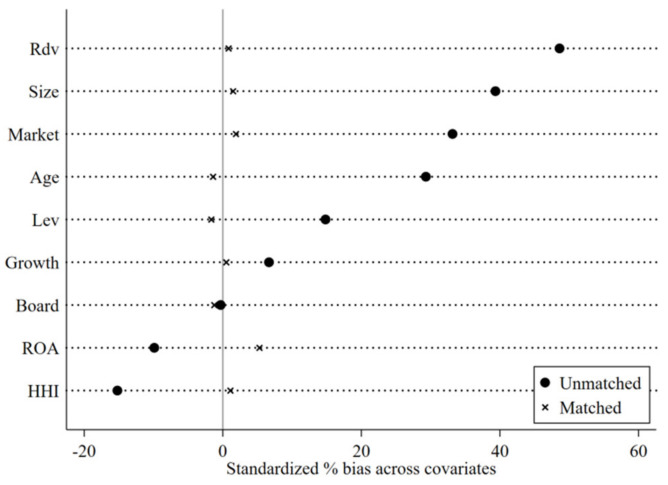
Standard deviation comparison before and after matching.

**Figure 3 ijerph-20-01840-f003:**
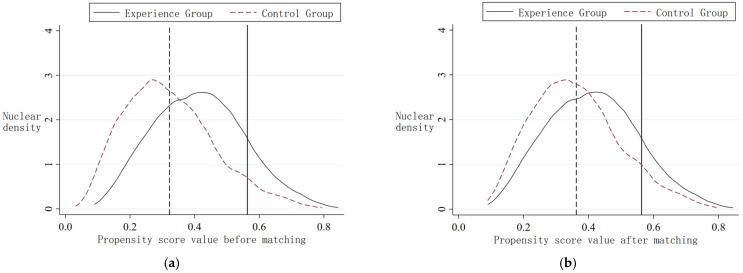
(**a**) Propensity score value before matching. (**b**) Propensity score value after matching.

**Table 1 ijerph-20-01840-t001:** Descriptive statistics for the variables.

Variables	*n*	Mean	SD	Min	Max
*GT*	6918	6.6811	6.527	1.00	28.00
*XGT*	6918	4.7615	4.078	1.00	17.00
*SGT*	6918	1.9079	2.736	0.00	11.00
*Size*	6918	21.8825	1.072	20.01	25.05
*Age*	6918	2.8089	0.335	1.79	3.40
*Rdv*	6918	17.6815	1.303	13.91	21.01
*Lev*	6918	0.3788	0.193	0.05	0.85
*ROA*	6918	0.0457	0.058	−0.17	0.20
*Growth*	6918	0.1498	0.325	−0.43	1.82
*Board*	6918	2.1185	0.187	1.61	2.56
*Market*	6918	8.5030	1.719	4.10	10.96
*HHI*	6918	0.4037	0.374	0.00	1.00

**Table 2 ijerph-20-01840-t002:** Balance test results.

Variable		Mean		% Reduct		*t*-Test	
		Treated	Control	% Bias	|Bias|	*t*	*p* > |*t*|
*Size*	U	22.15	21.736	39.3		15.64	0.000
	M	22.141	22.125	1.5	96.3	0.50	0.620
*Age*	U	2.8706	2.775	29.3		11.46	0.000
	M	2.8692	2.8738	−1.4	95.2	−0.53	0.600
*Rdv*	U	18.087	17.464	48.6		19.24	0.000
	M	18.069	18.059	0.8	98.3	0.30	0.768
*Lev*	U	0.39698	0.36885	14.8		5.82	0.000
	M	0.39628	0.39944	−1.7	88.7	−0.58	0.559
*ROA*	U	0.04204	0.04772	−9.9		−3.90	0.000
	M	0.04212	0.03908	5.3	46.6	1.79	0.073
*Growth*	U	0.16366	0.14216	6.7		2.64	0.008
	M	0.16358	0.16197	0.5	92.5	0.17	0.869
*Board*	U	2.1181	2.1187	−0.3		−0.12	0.902
	M	2.118	2.1203	−1.2	−304.2	−0.43	0.667
*Market*	U	8.8628	8.3052	33.2		13.07	0.000
	M	8.8572	8.8254	1.9	94.3	0.69	0.489
*HHI*	U	0.36752	0.42364	−15.2		−5.98	0.000
	M	0.36833	0.36433	1.1	92.9	0.39	0.696

**Table 3 ijerph-20-01840-t003:** DID results.

Variable	(1)	(2)	(3)	(4)
	GT	GT	XGT	SGT
*Post* × *Treat*	1.0572 *** (6.7828)	0.3176 ** (2.3100)	0.2617 *** (3.0634)	0.0575 (0.8751)
*Size*		0.3687 *** (2.5843)	0.2818 *** (3.1773)	0.0813 (1.1923)
*Age*		−2.5601 *** (−2.9852)	−2.1392 *** (−4.0135)	−0.3950 (−0.9642)
*Rdv*		0.0580 (0.6569)	0.0238 (0.4330)	0.0308 (0.7292)
*Lev*		0.1220 (0.2524)	0.3419 (1.1379)	−0.1907 (−0.8259)
*ROA*		0.7421 (0.7690)	0.3142 (0.5239)	0.4311 (0.9353)
*Growth*		−0.4293 *** (−3.6450)	−0.3320 *** (−4.5346)	−0.1092 * (−1.9405)
*Board*		−0.5838 (−1.4732)	−0.3328 (−1.3512)	−0.2087 (−1.1027)
*Market*		0.2261 ** (2.0717)	0.0874 (1.2892)	0.1321 ** (2.5352)
*HHI*		−1.2454 *** (−9.0873)	−0.4763 *** (−5.5919)	−0.7549 *** (−11.5314)
*Year*	NO	YES	YES	YES
*Enterprise*	NO	YES	YES	YES
*Constant*	6.8074 *** (87.3481)	4.8221 (1.3134)	4.4206 * (1.9372)	0.4550 (0.2594)
*n*	9784	9749	9749	9749
adj. *R*^2^	0.005	0.772	0.773	0.707

Note: *, **, and *** mean *p* < 0.1, 0.05, and 0.01 respectively. The standard error in brackets is robust.

**Table 4 ijerph-20-01840-t004:** 1:2 Nearest Neighbor Matching Balance Test Results.

Variable		Mean		% Reduct		*t*-Test	
		Treated	Control	% Bias	|Bias|	*t*	*p* > |*t*|
*Size*	U	22.15	21.736	39.3		15.64	0.000
	M	22.141	22.129	1.1	97.1	0.38	0.703
*Age*	U	2.8706	2.775	29.3		11.46	0.000
	M	2.8692	2.8716	−0.7	97.5	−0.27	0.784
*Rdv*	U	18.069	17.464	48.6		19.24	0.000
	M	18.069	18.07	−0.1	99.9	−0.03	0.979
*Lev*	U	0.39698	0.36885	14.8		5.82	0.000
	M	0.39628	0.39949	−1.7	88.6	−0.59	0.553
*ROA*	U	0.04204	0.04772	−9.9		−3.90	0.000
	M	0.04212	0.03915	5.2	47.8	1.76	0.079
*Growth*	U	0.16366	0.14216	6.7		2.64	0.008
	M	0.16358	0.17027	−2.1	68.9	−0.69	0.491
*Board*	U	2.1181	2.1187	−0.3		−0.12	0.902
	M	2.118	2.1224	−2.4	−679.9	−0.84	0.402
*Market*	U	8.8628	8.3052	33.2		13.07	0.000
	M	8.8572	8.8187	2.3	93.1	0.84	0.402
*HHI*	U	0.36752	0.42364	−15.2		−5.98	0.000
	M	0.36833	0.37045	−0.6	96.2	−0.21	0.937

**Table 5 ijerph-20-01840-t005:** Stability test results of DID regression.

Variable	(1)	(2)	(3)	(4)	(5)	(6)
	GT	GT	XGT	SGT	GT	GT
*Post* × *Treat*	1.0660 *** (6.4067)	0.2730 * (1.7225)	0.2347 ** (2.3804)	0.0369 (0.4843)	0.5562 * (1.7746)	0.3285 ** (2.2458)
*Size*		0.4328 ** (2.5751)	0.3374 *** (3.2265)	0.0968 (1.1988)	1.9852 *** (7.8428)	0.3442 ** (2.1866)
*Age*		−2.2960 ** (−2.2987)	−2.1574 *** (−3.4712)	−0.1015 (−0.2113)	0.8318 * (1.6929)	−3.4495 *** (−3.6220)
*Rdv*		−0.0185 (−0.1744)	−0.0417 (−0.6310)	0.0105 (0.2049)	−0.0571 (−0.3096)	0.1480 (1.5879)
*Lev*		−0.2002 (−0.3548)	0.1535 (0.4371)	−0.3239 (−1.1938)	−0.1093 (−0.1119)	−0.1486 (−0.2853)
*ROA*		0.1245 (0.1096)	−0.1115 (−0.1578)	0.2318 (0.4247)	7.9524 *** (3.6871)	0.6809 (0.6394)
*Growth*		−0.2926 ** (−2.0897)	−0.2636 *** (−3.0256)	−0.0464 (−0.6893)	−1.7190 *** (−7.2831)	−0.3347 *** (−2.5822)
*Board*		−0.5813 (−1.2376)	−0.2737 (−0.9365)	−0.2636 (−1.1678)	2.2633 *** (2.9098)	−0.8767 ** (−2.0363)
*Market*		0.0694 (0.5344)	0.0039 (0.0486)	0.0596 (0.9551)	−0.3256 *** (−3.2238)	0.3191 *** (2.7241)
*HHI*		−1.0091 *** (−6.3647)	−0.3410 *** (−3.4561)	−0.6594 *** (−8.6527)	−0.2663 (−0.8298)	−1.4837 *** (−10.1423)
*Year*	NO	YES	YES	YES	YES	YES
*Enterprise*	NO	YES	YES	YES	YES	YES
*Constant*	6.7987 *** (70.7699)	5.5229 (1.3078)	5.1110 * (1.9450)	0.4354 (0.2145)	−42.0946 *** (−10.1171)	6.5055 (1.6146)
*n*	7338	7282	7282	7282	6918	8526
adj. *R*^2^	0.005	0.770	0.770	0.702	0.196	0.779

Note: *, **, and *** mean *p* < 0.1, 0.05, and 0.01 respectively. The standard error in brackets is robust.

**Table 6 ijerph-20-01840-t006:** The results of the internal mechanism.

Variable	(1)	(2)	(3)	(4)	(5)	(6)
	Dig	GT	GI	GT	RR	GT
*Post* × *Treat*	0.0051 ** (2.1410)	0.3295 ** (2.3979)	0.1594 ** (2.1960)	0.3099 ** (2.2544)	0.0989 *** (2.7297)	0.3048 ** (2.2175)
*Dig*		−2.3591 *** (−3.7895)				
*GI*				0.0480 ** (2.3707)		
*RR*						0.1289 *** (3.1801)
*Size*	−0.0005 (−0.2023)	0.3676 *** (2.5780)	0.1629 ** (2.1615)	0.3609 ** (2.5295)	0.2090 *** (5.5570)	0.3418 ** (2.3925)
*Age*	−0.0454 *** (−3.0856)	−2.6672 *** (−3.1108)	−1.7939 *** (−3.9608)	−2.4740 *** (−2.8830)	−0.6101 *** (−2.6988)	−2.4814 *** (−2.8938)
*Rdv*	0.0052 *** (3.4509)	0.0703 (0.7966)	0.0792* (1.6973)	0.0542 (0.6139)	−0.1507 *** (−6.4731)	0.0774 (0.8752)
*Lev*	−0.0243 *** (−2.9336)	0.0646 (0.1337)	1.2447 *** (4.8750)	0.0623 (0.1288)	−6.8979 *** (−54.1307)	1.0112 * (1.8114)
*ROA*	−0.0238 (−1.4398)	0.6858 (0.7111)	1.3459 *** (2.6408)	0.6775 (0.7020)	−0.9542 *** (−3.7514)	0.8651 (0.8962)
*Growth*	0.0033 (1.6401)	−0.4215 *** (−3.5808)	−0.2942 *** (−4.7293)	−0.4152 *** (−3.5216)	−0.3029 *** (−9.7570)	−0.3903 *** (−3.2973)
*Board*	0.0113 * (1.6610)	−0.5572 (−1.4068)	−0.9253 *** (−4.4212)	−0.5394 (−1.3600)	−0.0068 (−0.0655)	−0.5829 (−1.4717)
*Market*	−0.0047 ** (−2.5296)	0.2149 ** (1.9701)	0.1643 *** (2.8511)	0.2182 ** (1.9991)	−0.0378 (−1.3134)	0.2309 ** (2.1172)
*HHI*	0.0020 (0.8553)	−1.2406 *** (−9.0592)	0.1582 ** (2.1854)	−1.2530 *** (−9.1426)	0.0178 (0.4922)	−1.2477 *** (−9.1087)
*Year*	YES	YES	YES	YES	YES	YES
*Enterprise*	YES	YES	YES	YES	YES	YES
*Constant*	0.1152 * (1.8286)	5.0939 (1.3882)	1.3102 (0.6757)	4.7593 (1.2965)	4.8372 *** (4.9981)	4.1986 (1.1425)
*n*	9749	9749	9749	9749	9749	9749
adj. *R*^2^	0.681	0.773	0.783	0.772	0.820	0.773

Note: *, **, and *** mean *p* < 0.1, 0.05, and 0.01 respectively. The standard error in brackets is robust.

**Table 7 ijerph-20-01840-t007:** The results of heterogeneity effects.

Variable	The Internal Governance				The External Environment			
	(1)	(2)	(3)	(4)	(5)	(6)	(7)	(8)
	State-Owned Enterprise	Non-State-Owned Enterprises	High Level of Corporate Governance	Low Level of Corporate Governance	Heavily Polluting Enterprises	Non-heavily Polluting Enterprises	Eastern Region	Central and Western Regions
*Post* × *Treat*	0.1257 (0.4457)	0.3414 ** (2.1560)	0.5623 *** (3.1018)	0.0175 (0.0834)	−7.7776 *** (−2.4 × 10^13^)	0.3434 ** (2.4960)	0.4392 *** (2.9447)	−0.1715 (−0.5245)
*Size*	0.1100 (0.3605)	0.3528 ** (2.1146)	0.4531 ** (2.2775)	0.2669 (1.1123)	19.0061 *** (4.0 × 10^13^)	0.3255 ** (2.2779)	0.2951 * (1.8932)	0.9937 *** (2.8821)
*Age*	−2.4997 (−1.0867)	−2.4938 *** (−2.5917)	2.6159 ** (2.2784)	−2.0037 (−1.4258)	−27.6934 *** (−4.7 × 10^12^)	−2.4530 *** (−2.8602)	−2.4404 *** (−2.7801)	0.2336 (0.0849)
*Rdv*	−0.3734 ** (−2.4259)	0.2687 ** (2.4295)	0.3355 *** (2.5949)	−0.1470 (−1.1197)	−18.7216 *** (−5.1 × 10^13^)	0.0785 (0.8880)	−0.0400 (−0.3878)	0.0613 (0.3430)
*Lev*	0.1490 (0.1345)	0.1746 (0.3191)	1.0906 * (1.7169)	−0.3013 (−0.3629)	56.8975 *** (2.4 × 10^13^)	0.0946 (0.1956)	0.1276 (0.2475)	0.4824 (0.3932)
*ROA*	5.7505 ** (2.2998)	0.1337 (0.1274)	1.8088 (1.5286)	−3.0122 * (−1.8252)	−21.9108 *** (−1.1 × 10^13^)	0.8915 (0.9224)	−0.1567 (−0.1542)	3.1901 (1.2267)
*Growth*	−0.2748 (−1.1508)	−0.4957 *** (−3.6317)	−0.2403 (−1.4815)	−0.2865 * (−1.6619)	−4.0341 *** (−2.3 × 10^13^)	−0.4951 *** (−4.1547)	−0.4836 *** (−3.7146)	−0.2622 (−0.9692)
*Board*	0.1441 (0.1739)	−0.9025 ** (−1.9738)	−2.0514 *** (−3.9586)	−0.5076 (−0.6680)	−41.8071 *** (−2.7 × 10^13^)	−0.4524 (−1.1386)	−0.6067 (−1.3970)	−0.5256 (−0.5708)
*Market*	0.3736 (1.6032)	0.2011 (1.6163)	−0.0261 (−0.1792)	0.5317 *** (3.1138)	−62.8191 *** (−3.8 × 10^13^)	0.2364 ** (2.1662)	0.2581 ** (2.2588)	−0.2862 (−0.7154)
*HHI*	−1.0691 *** (−4.0798)	−1.3158 *** (−8.0520)	−1.4708 *** (−7.6416)	−1.0112 *** (−5.0715)	2.8681 *** (8.9 × 10^12^)	−1.2492 *** (−9.1038)	−1.2104 *** (−8.2280)	−1.3158 *** (−3.8651)
*Year*	YES	YES	YES	YES	YES	YES	YES	YES
*Enterprise*	YES	YES	YES	YES	YES	YES	YES	YES
*Constant*	17.4366 * (1.7786)	1.4323 (0.3447)	−13.1668 *** (−2.6790)	8.0882 (1.2948)	617.4659 *** (4.1 × 10^13^)	4.7024 (1.2807)	7.0860 * (1.8310)	−12.2689 (−1.1158)
*n*	2601	7136	4593	5059	64	9685	7601	2142
adj. *R*^2^	0.792	0.759	0.777	0.785	1.000	0.771	0.777	0.758

Note: *, **, and *** mean *p* < 0.1, 0.05, and 0.01 respectively. The standard error in brackets is robust.

## Data Availability

Not applicable.

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
