# Peer review of "Impacts of Digital Information Management Systems on Green Transformation of Manufacturing Enterprises"

_ijerph, 2023, doi:10.3390/ijerph20031840_

Round 1
Reviewer 1 Report
The article Impacts of Digital Information Management System on Green Transformation of Manufacturing Enterprises aimed to explore its impact on the green transformation of manufacturing enterprises and its mechanism. The research article is well written and provide extention in body of knowledge. I beleive the article can be published after minor corrections. I have following suggestions for improvenment.
1. Introduction needs to highlight the novelty of the research.
2. The digital transformation was attributed to the three important components e.g .ERP, MES, PCS. However, all these components are internal to the organization and data flowing through the organization. However, digital transforation is beyond the organizational boundries.
3. Development of the framework is not clear, How reserchers reached to the hypothesis? whats other important constructs were ignored and why?
4. Statistical analysis is correct and strengthen the study. Findings are in line with the conclusions.
Reviewer 2 Report
This manuscript is basically acceptable.
The logic of Figure 2 is too ideal and it should be removed or further refined.
Reviewer 3 Report
Dear authors
My personal opinion is that after reading the paper, the manuscript is of potential interest to the readership of this journal, but there are issues that must be addressed:
In general:
1. Background – Expand a little more to highlight the research problem to highlight the study's need.
2. Methodology - expand a little more.
3. Findings: Should align with the study goal.
Introduction
A concise introduction to enable the reader's understanding of the research problem.
• The introduction should clearly illustrate (1) what we know (the key theoretical perspectives and empirical findings) and what do we not know (major, unaddressed puzzle, controversy, or paradox does the study addresses, or why it needs to be addressed and why this matters). And, (2) what will we learn from the study and how does the study fundamentally change, challenge, or advance scholars’ understanding. Much sharper problematization is required so that the introduction draws the reader into the paper. The introduction therefore needs to do a better job in setting the stage for the articulation of the theoretical contributions of the study. At the end of the introduction, we should have a clear idea of what the paper is about (i.e. its motivation, the gap in understanding that the paper is trying to address and summary of theoretical contributions).
Literature review
The paper should relate coherently and convincingly with issues of real-world significance. This is a crucial phase contributing to research design.
Suggestions
• Add more information to enable readers' understanding of the authors' view.
Methodology
The method should be adequately described to show how the research was conducted to improve clarity and transparency.
Findings and discussion
Needs clear and comprehensive explanations to assist readers' understanding.
Conclusion
The conclusion falls short of providing sufficient information that would allow a reader to understand the contribution of this research. What was found?
Reference.
Authors should use more recent research.
- Using the following reference could be beneficial as these add more evidence to the literature review section:
Knowledge sharing and achieving competitive advantage in international environments: The case of Iranian digital start-ups. In International Entrepreneurship in Emerging Markets (pp. 206-224). Routledge.
Best of luck with the further development of the paper.
Reviewer 4 Report
Dear Authors,
This paper takes the adoption of digital information management system by China’s enterprises as a "quasi natural" experiment, and uses PSM-DID to explore its impact on the green transformation of manufacturing enterprises and its mechanism. Such study has important inspiration for Chinese manufacturing enterprises to use digitalization capabilities to seek green sustainable development under the wave of digital economy development.
Point 1, There are many practical operational problems in manufacturing enterprises(Line76-Line82), but these problems do not seem to be deeply related to the green transformation development discussed in this paper. There should be a transition statements from these practical problems to the green transformation.
Point 2, There exists “R&D”in Line102, is it equal to “R&D”? Please check it ou.
Point 3, Is the statements of “double carbon” goal clearly? Or whether you should write it separately?
Point 4, What is the relationship between the three hypotheses and the three sub-components of the digital information management system(ERP MES PCS)? If it matters, please explain in detail. If not, why are they connected by arrows in Figure 2?
Point 5, The core explanatory variable of DID model is the interaction term of two dummy variables, namely group dummy variable and time dummy variable. According to the data description, panel data is used in this article, but this model sets the time dummy variable as posti,t not postt, please check and explain clearly.
Point 6, The description of GT and the calculation process are a little unclear and complex. The author should add a clear explanation, such as showing the corresponding formula.
Point 7, The two pictures of Figure 4 lack horizontal headings, please add it.
Point 8, In the part of 4.3. Robustness Check, the model(3) and (4) choose XGT and SGT to be the explained variables, but the corresponding text does not give us reasons in detail? Please add explanations.
Point 9, The second part of 5.2. Heterogeneity Analysis chose corporate governance level to carry out heterogeneity analysis, but which attribute value of the enterprise data is used to clarify the level, it seems the paper does’nt give us. Can the authors show us?
Point 10, PSM-DID is generally used in the robustness test of traditional DID analysis, but this paper uses it directly as a core analytics tool, so what is the author's purpose for doing so, can authors explain it? Meanwhile, robustness tests should include placebo tests and counterfactual tests. Why is this part not included in the article? Lastly, parallel trend test is a necessary assumption premise for DID analysis, but this part is not included in the paper. Please complete it.
Point 11, There are also some normative issues in writing, such as capitalization and punctuation. Please review and revise carefully.
